# Can the Frailty of Older Adults in China Change? Evidence from a Random-Intercept Latent Transition Profile Analysis

**DOI:** 10.3390/bs13090723

**Published:** 2023-08-30

**Authors:** Guangming Li, Yuxi Pan

**Affiliations:** 1Key Laboratory of Brain, Cognition and Education Sciences, Ministry of Education, South China Normal University, Guangzhou 510631, China; 2School of Psychology, Center for Studies of Psychological Application and Guangdong Key Laboratory of Mental Health and Cognitive Science, South China Normal University, Guangzhou 510631, China

**Keywords:** older adults, frailty, self-reported health, social function, mental health, cognitive function, functional limitation, morbidity status, random-intercept latent transition profile analysis (RI-LTPA), longitudinal study

## Abstract

Background: A major aspect of caring for older adults in the medical field is addressing their health risks. The term “frailty” is generally used to describe the changes in health risks of older adults. Although there is considerable heterogeneity in the Chinese older adult population who are classified as frail, there remain few relevant studies. Furthermore, there is a lack of research on the frailty status transitions of older Chinese adults at different time points. This research intends to determine the frailty status and category of older adults according to their physical, psychological, social, and cognitive function domains, and on this basis, to investigate changes in their frailty states. Methods: This article studied 2791 respondents who were over 60 years old (*n* = 2791; 53.2% were women) from the Chinese Longitudinal Healthy Longevity Survey (CLHLS) follow-up survey on factors affecting the health of older adults in China. In this article, the frailty variables include self-reported health, social function, mental health, cognitive function, functional limitations, and morbidity status. Random-intercept latent transition profile analysis (RI-LTPA) was used to divide older adults into different subgroups, and then an in-depth analysis of the state transitions was conducted. Results: The latent profile analysis revealed that the evaluation results of the frailty state of older adults showed obvious group heterogeneity. Each fitting index supported four latent states, which were named according to the degree of the symptoms (i.e., multi-frailty, severe socially frailty, mild socially frailty, and relatively healthy frailty). Based on the categorical probability and the probability of transition, it can be concluded that most of the samples belonged to the healthy population, and the health status had generally improved across the four time points. The relatively healthy frailty group and the severe socially frailty group have relatively strong stabilities. The multi-frailty group and the mild socially frailty group had the highest probability of joining to the relatively healthy frailty group. Strengthening social interactions among older adults and promoting their participation in social activities can significantly improve their frail state. Conclusions: This study supplements related research on frailty. Firstly, it deepens the meaning of frailty, which is defined based on four aspects: physical, psychological, cognitive, and social functioning. Secondly, it divides frailty into different sub-categories. Frailty is discussed from the perspective of longitudinal research, which can provide practical adjustment suggestions for older adult nursing intervention systems and measures in China.

## 1. Introduction

The aging of the population and the care of older adults pose major challenges to the global health care system. In 2020, the number of people aged 60 and over around the world surpassed the number of children under 5 years old for the first time in history [1]. China’s aging problem is also very severe. According to the seventh census, the population over 60 is currently 264,018,800, accounting for 18.7%, and the population over 65 is 13.5%. These statistics characterize the national conditions of population aging as having the “three most” characteristics: the largest number, the fastest speed, and the heaviest response task. This shows that China has entered an aging society and is facing more severe new challenges. Therefore, a further deepening of the research on the aging population is urgently needed [2].

At the same time, due to the rapid aging of the population and the increase in the proportion of older adults each year, the proportion of China’s frail older adults who have lost a part or all of their ability to take care of themselves is increasing. In this growing group, the heterogeneity is large and their health conditions are very different, leading to different care needs [3]. Most intervention strategies target different age groups, but the process of aging among individuals is different. If the population is simply distinguished by age, this will likely result in significant increases in nursing costs and inefficiencies. Therefore, a more precise assessment of older adults is needed to promote the formulation of individualized intervention programs.

Frailty was first characterized in the late 1960s [4]. A cross-sectional study of older adults in the community described frailty for the first time; that is, a disproportionate response to adverse events. Subsequently, frailty was gradually used to evaluate the health status of older adults. The concept of frailty was formally put forward at the 1978 American Federal Conference for older adults. Rockwood et al. [5] used the frailty dynamic model to better explain the concept of frailty. This model describes health and disease in older adults as two aspects of the same phenomenon. The health aspect is an asset, and the disease aspect is a deficit. The balance of these two aspects determines whether an individual can live in the community without relying on others. Woodhouse [6] defined frail older adults as those who are over 65 years old and cannot take care of themselves in daily life, relying on the care of others or needing to enter medical rehabilitation institutions to obtain care. There are also scholars that argue that frailty in older adults is an intermediate state between a lack of health and non-serious damage. However, this intermediate state is different from the sub-health state of young and middle-aged populations. Most frail older adults have a series of chronic diseases, and some degree of their frailty may be an acute event or a serious disease (osteoporosis, infection, malignant tumor, depression, etc.) As a consequence, older adults are frail before death [7]. Therefore, frailty is not only related to age and aging, but is also closely related to disease. Furthermore, Fried et al. [8] pioneered the definition of frailty, which attracted the attention of professionals in the field of older adults. This resulted in a rapid increase in global research on frailty. Frailty has gradually proven to be a favorable indicator of health status and care needs. It has been widely used in the fields of public health, medicine, nursing, psychology, sociology, and demography. It represents core knowledge and a key teaching point in the field of aging. In 2004, the US National Conference on Elders identified research in the field of frailty as an important task for improving the quality of health care in the country in the future. In 2007, the United States discussed frailty and its relevance to nursing practice at the Hartford Aged Care Institution. Senior practice nurses should have the ability to address a variety of chronic health problems and manage the mental and physical frailty of older adults [9].

The current definition of frailty is based on multiple measures. Frailty has multiple manifestations, and no single symptom is sufficient or necessary. Manifestations include appearance (consistent with or inconsistent with age), nutritional status (lean, weight loss), subjective health ratings (health perceptions), performance (cognition, fatigue), sensory/physical impairment (vision, hearing, strength), and current care (medicine, hospital) [10]. It can be said that frailty is a group of clinical syndromes, which is not synonymous with sickness or disability. It is caused by degenerative changes in the body and a series of chronic diseases. Frailty is not only related to age and aging, but also closely related to disease. It is often used to describe chronic health problems in older adults over 65 years old, especially among older adults over 80 years [11,12]. The definition and measurement of frailty are becoming more and more complete, but there are currently fewer studies that analyze the heterogeneity of frailty, for example, Looman et al. [13]. In fact, latent transition profile analyses of the frailty states of older adults should be used, as on the basis of previous studies, “frailty” can be indexed and classified in profile, and then different aging processes can be distinguished so as to provide a theoretical basis for personalized nursing. Therefore, this study integrates the previous literature and comprehensively measures the state of frailty on this basis, not only based on functional limitations, the decline in the quality of daily living, and frequent diseases, but also based on psychological, cognitive, and social functions.

Given the feature of frailty, several relevant questions are naturally raised: (a) Do distinct frailty states exist? (b) What is the pattern of transition from one state (e.g., addiction) to another? (c) What are the effects of some characteristics of older adults on frailty state? Random-intercept latent transition analysis (RI-LTA) perfectly accommodates the three concerns. The following hypotheses are proposed:
**Hypothesis** **1.***There is heterogeneity in the frail population of older adults, and there are different types of frail populations of older adults with varying frail states*.
**Hypothesis** **2.***Across different time points (T1, T2, T3, T4), the frailty state of older adults can transform mutually, and the transformation has a certain pattern (or patterns)*.
**Hypothesis** **3.***Strengthening social interaction among older adults and promoting their participation in social activities is beneficial for the development of their frail state toward health*.

Latent transition analysis (LTA) is based on the latent Markov model of latent class analysis [14]. At present, the LTA model is relatively widely used in psychology, mainly in the following three scenarios: developmental psychology [15], to explore children’s dyslexia [16], mental health education [17,18,19,20,21], clinical psychology, and medical fields [22,23,24,25].

The mathematical model used in this study is the optimization model of the latent transition analysis model, namely the random-intercept latent transition analysis model. Compared with regular LTA, RI-LTA can not only study longitudinal heterogeneity, but also perform two-level analyses, that is, the self-transition generated across time is analyzed at level 1, and the difference among individuals that are constant across time is analyzed at level 2 [26].

The objective of the study is to explore the types of frailty among older adults and their transformation mechanisms in order to provide reference for improving the health level of older adults. According to the examples and Monte Carlo simulation studies in Muthén and Asparouhov [26], RI-LTA can present data better than regular LTA, and more accurately assess the variability and stability of mental state over time. This new model has changed the interpretation of the process of psychological change. Therefore, this study intends to further optimize the model on the basis of the previous literature using the random-intercept latent transition profile analysis (RI-LTPA) to conduct research on the transition of the frailty state of older adults and provide more meaningful enlightenment for the actual nursing work.

## 2. Methods

### 2.1. Sample

Data were derived from the Chinese Longitudinal Healthy Longevity Survey (CLHLS) [1], which is a large national representative database focusing on older adults in China, covering 631 county-level administrative regions in 23 provinces, autonomous regions, and municipalities in China. Participants included a large, random sample of Chinese elders involved in the CLHLS [27]. The design type of this study is longitudinal research. The CLHLS data were collected at seven waves over 16 years, first in 1998, and then in 2000, 2002, 2005, 2008, 2011, and 2014. The CLHLS examined Chinese elders’ health conditions, everyday functioning, self-perceptions of health status and quality of life, life satisfaction, mental attitude, and feelings about aging [27].

The survey content involves the basic conditions, social activities, mental status, economic sources and other modules of the individual and family of older adults. This study mainly uses data from four of the tracking stages: 2005, 2008, 2011, and 2014. The initial sample was 15,638. Interviews started in 2005. Excluding older adults who died or were lost to follow-up, there were a total of 2791 respondents with an average age of 75.

The following exclusion criteria were established: (1) cases that were missing two or more times and (2) subjects with severe missing data parts of a certain scale. To indicate whether the data were randomly missing or not, cross-tab chi-squared tests were performed on the selected data, and the chi-squared value was not significant, χ^2^(*df*) = 35.708, *p* = 0.406, indicating that the missing data were missing at random [1]. This study included a sample of 2791 older adults from 64 to 108 years old, of which 1484 were women, accounting for 53.2% of the total; 1404 were rural, accounting for 50.3% of the total; and 2649 were 64–90 years old, accounting for 94.9% of the total.

### 2.2. Measurement

#### 2.2.1. Self-Reported Health

Self-reported health is evaluated with two items from RAND-36 (see Appendix A) [27]. The first item allows older adults to evaluate their own current general health in the following answer categories: excellent; very good; good; fair; poor. The second item is self-reported health compared to 1 year ago: much better; somewhat better; about the same; somewhat worse; much worse. With reverse coding, the higher the score is, the higher the self-reported health is. The Cronbach’s alpha coefficients of the self-reported health scale were 0.782 at T1 (2005), 0.799 at T2 (2008), 0.784 at T3 (2011), and 0.799 at T4 (2014).

#### 2.2.2. Social Function

Social functions mainly include four variables: marital status, living style, social activities, and availability of help (see Appendix A) [28]. Among them, social activities mainly include 9 specific activities, which are expressed in an aggregated form. The higher the score is, the higher the social function. The Cronbach’s alpha coefficients of the social function scale were 0.812 at T1 (2005), 0.805 at T2 (2008), 0.824 at T3 (2011), and 0.809 at T4 (2014).

#### 2.2.3. Mental Health

Mental health indicators include the evaluation of the status quo and personality and emotional characteristics (see Appendix A) [29]. The evaluation items for the status quo include: “How do you feel about your current life?”, “How do you feel about your own health now?”, and “Have you felt your health status has changed in the past year?”. Questions on personality and emotional characteristics include: “Are you able to think about anything you encounter?”, “Do you like to keep things clean and tidy?”, “Do you feel energetic?”, “Do you feel ashamed, regret, or guilty about what you have done?”, “Are you angry because you can’t understand the people or things around you?”, “Are you in charge of your own affairs?”, and “Do you often feel that the people around you are untrustworthy?”. Both parts are five-level scores. A decline in score value indicates increasingly negative emotions and worsening mental function. The Cronbach’s alpha coefficients of the meal health scale were 0.841 at T1 (2005), 0.802 at T2 (2008), 0.809 at T3 (2011), and 0.804 at T4 (2014).

#### 2.2.4. Cognitive Function

The measurement of cognitive function in this survey used the Mental State Examination (MSE) scale (see Appendix A) [30]. The initial test content of the MSE scale includes time orientation, location orientation, immediate memory, attention and calculation, recall, naming, retelling, 3 levels of instruction, reading comprehension, writing, and tracing. The maximum score is 30 points. The higher the value is, the better the cognitive function. In CLHLS, the MSE scale has been appropriately modified to adapt to China’s cultural environment. The scale has good reliability and an internal consistency coefficient of 0.89. In part C of the questionnaire, the items to measure the cognitive function of older adults include questions about general cognitive abilities of older adults, reaction ability, attention and calculation ability, memory and language comprehension, and coordination. Note that for question 6 (“The number of words spoken in one minute”), the maximum score is 7 points, while all other questions are 1 point, so the total score is still 30 points. In this study, a correct answer is counted as 1 point, an incorrect or unanswered answer is counted as 0 points, and the scores of 24 items are added to obtain the total cognitive function. The higher the score is, the higher the cognitive function is. The Cronbach’s alpha coefficients of the cognitive function scale were 0.832 at T1 (2005), 0.804 at T2 (2008), 0.811 at T3 (2011), and 0.809 at T4 (2014).

#### 2.2.5. Functional Limitation

Physiological health indicators mainly include functional limitation measurement and daily activity ability (activity of daily living, ADL; instrumental activity of daily living, IADL) (see Appendix A) [1]. Among them, there are five main functional restrictions, namely: “unable to put hand behind neck”, “unable to put hand behind lower neck”, “unable to raise arm upright”, “unable to stand up from sitting in a chair”, and “unable to pick up a book from floor”. The items of the ADL scale mainly include: “Do you need help from others when taking a bath (including scrubbing your upper or lower body)?”, “Do you need help from others when you dress (including finding and dressing)?”, “Do you use the toilet? Do you need help when urinating (including washing your hands after defecation, undressing and dressing, including using the toilet to urinate in the room)?”; “Do you need help from others when you are indoors? (Indoor activities refer to getting in and out of bed, sitting On a chair or stool or stand up from a chair or stool)?”, and “Can you control your bowel and urine?”. The IADL scale mainly includes seven items: “Can you visit your neighbor’s house alone?”, “Can you go out and buy things alone?”, “Can you cook alone if you need to? Can you wash clothes alone?”, “Can you walk 2 miles in a row?”, “Can you lift something weighing about 10 kg (5 kg)?”, “Can you squat and stand up three times in a row?”, and “Can you travel by public transport alone?”. All items are scored at three levels. The higher the score is, the worse the physiological function. The Cronbach’s alpha coefficients of the functional limitation scale were 0.822 at T1 (2005), 0.819 at T2 (2008), 0.829 at T3 (2011), and 0.815 at T4 (2014).

#### 2.2.6. Morbidity Status

Morbidity status is self-reported: participants could indicate their morbidities on a 24-item list of conditions (yes/no), such as hypertension, heart disease, and tuberculosis (see Appendix A) [13]. The higher the score is, the worse the morbidity status. The Cronbach’s alpha coefficients of the morbidity status scale were 0.832 at T1 (2005), 0.822 at T2 (2008), 0.854 at T3 (2011), and 0.849 at T4 (2014).

### 2.3. Data Processing

This study first used the $\chi^{2}$ test to compare the differences in the basic characteristics of older adults of different genders. Then, using the six dimensions of frailty indicators as explicit variables, LPA is used to explore the latent profiles of frailty. LPA mainly estimates the conditional probability (λ) of frailty and the probability of latent frailty profiles. The former describes the relationship between each latent frailty profile and the frailty indicators and serves as a reference value for the classification of the latent profiles of this study, while the latter reflects older adults’ distribution among latent frail classes.

Starting from the initial model, gradually increasing the number of classes, a total of 2 to 5 classes of LPA were established, using Bayesian information criterion (Bayesian information criterion, BIC) and sample-corrected Bayesian information criterion (adjusted Bayesian information criterion, aBIC), the indicators compared the fitting effects of each model and then the best model was selected. In this study, SPSS 26.0, and M*plus* 8.3 software were used for statistical analysis, and *p* < 0.05 was regarded as statistically significant. All these models were fit using M*plus* version 8.3 with Maximum Likelihood (ML) estimation. Missing data were processed using Full Information Maximum Likelihood (FIML) [31].

## 3. Results

### 3.1. Common Method Bias Test

According to Harman’s single-factor test [32], common method bias test was performed, and the unrotated principal component factor analysis was performed on the data of all measurement items at four time points. The results show that a total of three common factors with eigenvalues greater than 1 were extracted at time T1, and the variance explained by the first common factor was 32.737%. At T2, there were three common factors with eigenvalues greater than 1, and the first common factor, which was the variance explained by the factor, was 29.474%. At T3, there were two common factors with eigenvalues greater than 1, and the first common factor, which was the variance explained by the factor, was 32.130%. At T4, there were three common factors with eigenvalues greater than 1, and the first common factor—the variance explained by the factor—was 34.853%.

More than one common factor was extracted, and the interpretation rate of the first common factor was far less than the critical standard of 40%, suggesting that there is no obvious common method deviation in this study [33].

### 3.2. Determination of the Classes of Frailty Status for Older Adults

Table 1 shows the model fit statistics of the various class solutions. We chose the four-class solution, based on the significant LMR and the larger class proportion (the smallest class proportion is at least over 0.05).

### 3.3. Results of Frailty Status Classes for Older Adults

According to Figure 1, Figure 2, Figure 3 and Figure 4, the following figure shows the mean conditional probabilities of each category in the five aspects of frailty. Due to space limitations, only the conditional probabilities of the four time points are shown here.

Profile 1 (‘multi-frailty’, C1) is characterized by problems in seven items, especially functional limitation. They indicated having bad (mental) health and cognitive function. They were co-morbid, on average, and generally reported higher-than-average morbidities and more functional limitations.

Profile 2 (‘severe socially frailty’, C2) is characterized by suffering from serious problems in the social function domain. Compared with the other two older adults in the frailty profile, older adults in the third profile have no severe problems in the other six items except for severe social problems.

Profile 3 (‘mild socially frailty’, C3) is characterized by suffering from mild problems in the social function domain. Compared with the other two older adults in the frailty profile, older adults in the third profile have no serious problems in the other five items except for mild social problems.

Profile 4 (‘relatively healthy frailty’, C4) fundamentally differs from the other three profiles. Older adults in this profile were relatively healthy; they indicated having good (mental) health and had very few problems across all the domains. They were not co-morbid; on average, they generally reported lower-than-average morbidities and almost no functional limitations and social function problems.

### 3.4. Results of Frailty Status Transition for Older Adults

Table 2 shows the transition probability of the frailty state of older adults at T1 and T2. The diagonal of the transition matrix represents the probability that older adults remain in the initial state at the time points T1 and T2. The relatively healthy frailty group had the highest stability, with the probability of staying in the initial group reaching 80.2%, followed by the severe socially frailty group, with the probability of staying in the initial group reaching 60.7%. The multi-frailty group and the mild socially frailty group tended to transform to the relatively healthy frailty group, with the transition probability as high as 54.4% and 61.4%. Table 3 shows the transition probability of the frailty state of older adults at T2 and T3. Table 4 shows the transition probability of the frailty state of older adults at T3 and T4. Similarly, the relatively healthy frailty group and the severe socially frailty group have the highest probability of remaining in the initial state, and the multi-frailty group and the mild socially frailty group tend to change to the relatively healthy frailty group. In T1, T2, and T3, the transition probabilities of the mild socially frailty group to the other three profiles are the same, which is quite interesting and warrants further study.

## 4. Discussion

### 4.1. Analysis of Frailty Status Classes for Older Adults

In this study, based on the subjects’ three answers over 9 years, the latent profile analysis found that the evaluation results of the frailty state of older adults showed obvious group heterogeneity [34]. Each fitting index supports four latent states, which are named according to the degree of symptoms. In the relatively healthy group, the degree of frailty of all indicators is lower than the average level, while in the other three groups, more or less, all have higher-than-average frailty in some items. Among them, the relatively healthy frailty group in social function shows slight frailty, which indicates that some older adults have no major obstacles in physical function, cognitive function, and psychological function. The relatively healthy frailty group has good social function. The mild socially frailty group indicates that some older adults have no major obstacles in physical function, cognitive function, and psychological function, but have slight discomfort and incompetence in social function and interpersonal communication. Correspondingly, the severe socially frailty group indicated that some older adults have serious social function and interpersonal communication without other frailty problems [35]. The multi-frailty group shows higher-than-average frailty in all aspects, especially in morbidity and functional limit. Among them, the multi-frailty and the severe socially frailty group accounted for the least, while the mild socially frailty group and the relatively healthy frailty group accounted for more. This shows that older adults are in the partially frail group.

### 4.2. Analysis of Frailty Status Transition for Older Adults

In terms of latent transition probability, over time, the relatively healthy frailty group and the severe socially frailty group have relatively strong stability but overall indicate that the state of frailty is very stable for a certain period of time and has great invariability. The multi-frailty group and the mild socially frailty group have the highest probability of changing to the relatively healthy frailty group. On the one hand, the socially frail groups may change over time and strengthen their social communication function because of their changing mental capacities or better relationship with others. Not only is the transition probability of the frailty state of the mild socially frailty group high (0.614), but that of the severe socially frailty types are also relatively high (0.210~0.333), indicating that social factors such as social support, social activities, and social interaction cannot be ignored, which is consistent with some research (Li & Li, 2022; Bai et al., 2020) [1,28]. On the other hand, the multi-frailty group has the highest probability of changing to the relatively healthy frailty group. Over time, there may be improvement in the state of frailty, but there are more serious problems in the function of social communication [36]. Moreover, the probability of changing from the multi-frailty group to the mild socially frailty group is also relatively high (0.207~0.272), which indicates that older adults in the multi-frailty group may be able to perform some of the ability of daily living through some nursing interventions and alleviate the frailty problem [37,38].

### 4.3. The Innovation and Contribution of This Study

This research has shed some important light on actual medical care. First of all, this study uses a multi-faceted and multi-layered approach to define a comprehensive definition of frailty. On this basis, latent profile analysis is used to distinguish several frailty profiles, which proves that the frailty state of older adults is indeed heterogeneous. Secondly, previous studies have rarely analyzed the maintenance and transitions of the frailty state over time. This study uses a more advanced model (the random-intercept latent transition profile analysis model) to scientifically analyze the transition status of the four kinds of frailty. This can provide a theoretical basis for the focus of actual medical interventions, that is, at critical time points, four types of frailty among older adults in different states can be treated with different aspects of key care interventions. For example, for older adults in the severe socially frailty group, the focus is on improving their social communication skills and expanding their social activities [39].

### 4.4. Limitations

This study also has shortcomings. First, it did not include other covariates that may affect the frailty state on the basis of the model, such as living environment, education, marital status, socioeconomic status, etc., and so, the research did not achieve sufficient richness and completeness [1,40]. Secondly, because it was a large-scale questionnaire screening, the sample data’s missing rate was therefore relatively high, and the final sample size was only more than two thousand. It is difficult to use this as a representative of all older adults in China. Therefore, further research and expansion can be conducted on these two points in future research.

## 5. Conclusions

This paper showed that there were four types of frailty among older adults, namely multi-frailty group, severe socially frailty group, mild socially frailty group, and relatively healthy frailty group. During the measurement period of nine years, the frailty of the subjects showed a trend of improvement, but the proportions of the severe socially frailty group and the mild socially frailty group were relatively high at four time points. The mild socially frailty group could easily convert into the relatively healthy frailty group, but the severe socially frailty group was not relatively easily converted into the relatively healthy frailty group. But, if social factors are strengthened, both the mild socially frailty group and the severe socially frailty group may convert into the relatively healthy frailty group, which shows that performing well with regard to the mental health of the socially frail groups is key to prevent older adults’ frailty. A variety of psychological behavioral therapies can be integrated to prevent and intervene in their frailty among older adults.

Given the importance of social factors, in order to prevent the deterioration of the frailty of older adults, the government should build more places conducive to activities for older adults and actively encourage older adults to improve their physical activity level. Given our findings, public health interventions centered on increasing participation in social activities to enhance the activity of older adults should be endorsed on a national scale. Only in this way can frail older adults transform into healthy older adults.

## Figures and Tables

**Figure 1 behavsci-13-00723-f001:**
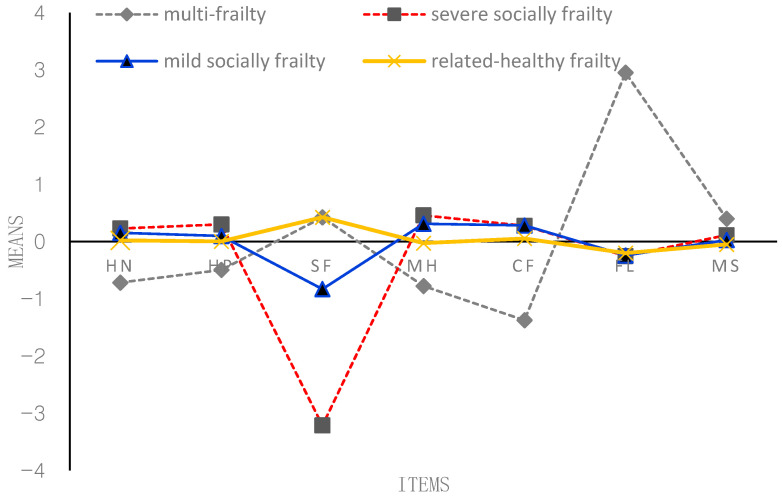
Conditional probability of each item of the frailty state at T1 (2005). *Note:* HN = self-reported health now; HP = self-reported health compared with past year; SF = social function; MH = mental health; CF = cognitive function; FL = functional limitation; MS = morbidity status. *The proportion of four profiles*: C1 = 6.84%; C2 = 6.99%; C3 = 13.44%; C4 = 72.73%.

**Figure 2 behavsci-13-00723-f002:**
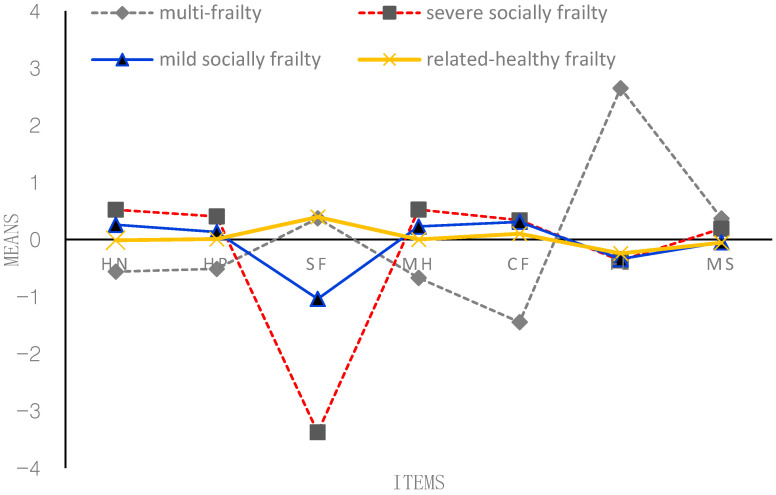
Conditional probability of each item of the frailty state at T2 (2008). *Note:* HN = self-reported health now; HP = self-reported health compared with past year; SF = social function; MH = mental health; CF = cognitive function; FL = functional limitation; MS = morbidity status. *The proportion of four profiles*: C1 = 8.89%; C2 = 6.20%; C3 = 11.18%; C4 = 73.73%.

**Figure 3 behavsci-13-00723-f003:**
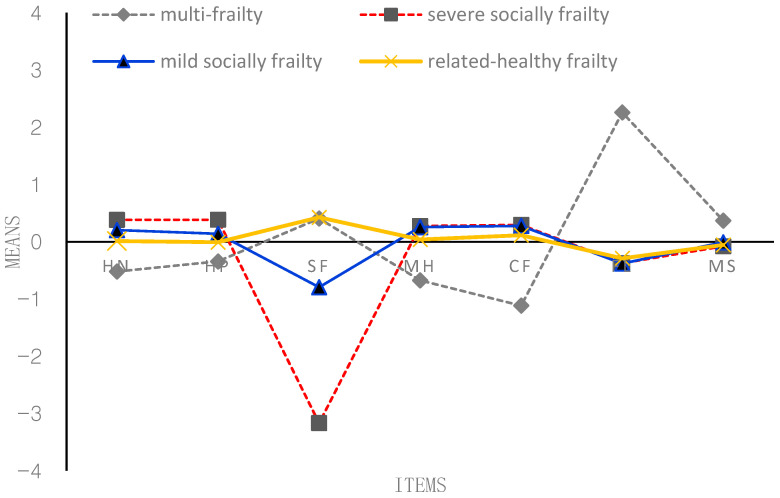
Conditional probability of each item of the frailty state at T3 (2011). *Note:* HN = self-reported health now; HP = self-reported health compared with past year; SF = social function; MH = mental health; CF = cognitive function; FL = functional limitation; MS = morbidity status. *The proportion of four profiles*: C1 = 11.93%; C2 = 7.27%; C3 = 13.40%; C4 = 67.40%.

**Figure 4 behavsci-13-00723-f004:**
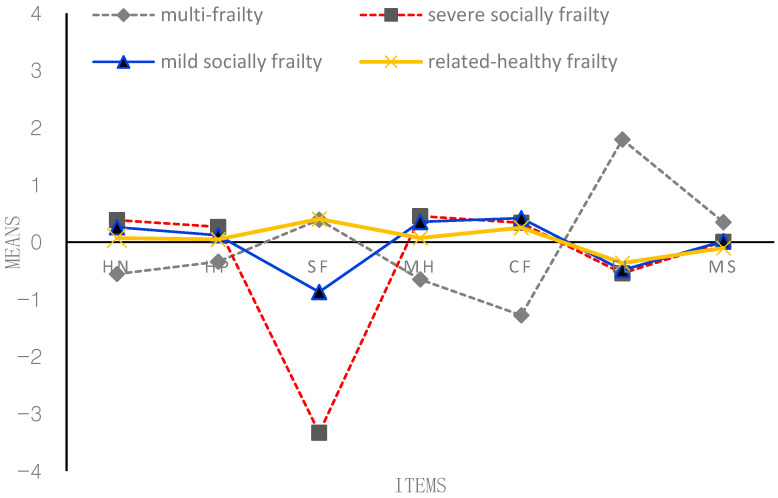
Conditional probability of each item of the frailty state at T4 (2014). *Note:* HN = self-reported health now; HP = self-reported health compared with past year; SF = social function; MH = mental health; CF = cognitive function; FL = functional limitation; MS = morbidity status. *The proportion of four profiles*: C1 = 6.63%; C2 = 12.15%; C3 = 18.05%; C4 = 63.17%.

**Table 1 behavsci-13-00723-t001:** Fitting index of latent category model of frailty state at T1, T2, T3, and T4.

Time	Class	Entropy	AIC	BIC	aBIC	LMRT	BLRT (*P*)	Class Proportion
2005, T1	2	0.998	52,210.459	52,341.011	52,271.109	0.0000	0.0000	0.89753/0.10247
	3	0.983	49,960.635	50,138.659	50,043.339	0.0006	0.0000	0.82623/0.10176/0.07202
	4	0.987	47,823.748	48,049.246	47,928.507	0.0000	0.0000	0.72734/0.13436/0.06987/0.06843
	5	0.976	47,079.206	47,352.177	47,206.019	0.4815	0.0000	0.70047/0.13436/0.06987/0.04407/0.05124
2008, T2	2	0.999	52,007.776	52,138.327	52,068.426	0.0000	0.0000	0.89579/0.10421
	3	0.981	49,296.588	49,474.612	49,379.292	0.0000	0.0000	0.80473/0.10391/0.09137
	4	0.985	47,125.369	47,350.867	47,230.128	0.0000	0.0000	0.73737/0.11179/0.08886/0.06198
	5	0.988	46,194.178	46,467.150	46,320.992	0.0002	0.0000	0.73737/0.11215/0.08850/0.03870/0.02329
2011, T3	2	0.997	52,125.415	52,255.966	52,186.065	0.0000	0.0000	0.89789/0.10211
	3	0.971	49,728.846	49,906.870	49,811.550	0.0000	0.0000	0.77463/0.12397/0.10140
	4	0.979	47,662.936	47,888.434	47,767.695	0.0000	0.0000	0.67395/0.13400/0.11931/0.07273
	5	0.849	48,153.711	48,426.682	48,280.524	0.0001	0.0000	0.53063/0.24328/0.10140/0.09137/0.03332
2014, T4	2	0.998	51,961.623	52,092.174	52,022.273	0.0000	0.0000	0.90426/0.09574
	3	0.960	48,925.019	49,103.043	49,007.723	0.0000	0.0000	0.71407/0.19149/0.09444
	4	0.969	46,689.901	46,915.399	46,794.660	0.0000	0.0000	0.63167/0.18058/0.12146/0.06628
	5	0.960	45,704.960	45,977.932	45,831.774	0.0000	0.0000	0.59441/0.16195/0.12110/0.06628/0.05625

Note: AIC = Akaike information criterion; BIC = Bayesian information criterion; aBIC = adjusted Bayesian information criterion; LMRT = Lo–Mendell–Rubin likelihood ratio test; BLRT = bootstrap likelihood ratio test.

**Table 2 behavsci-13-00723-t002:** Transition probability of profiles of frailty at T1 and T2.

		T2			
		Multi-Frailty	Severe Socially Frailty	Mild Socially Frailty	Relatively Healthy Frailty
T1	multi-frailty	0.195	0.045	0.215	0.544
	severe socially frailty	0.024	0.607	0.036	0.333
	mild socially frailty	0.105	0.060	0.220	0.614
	relatively healthy frailty	0.044	0.058	0.097	0.802

**Table 3 behavsci-13-00723-t003:** Transition probability of profiles of frailty at T2 and T3.

		T3			
		Multi-Frailty	Severe Socially Frailty	Mild Socially Frailty	Relatively Healthy Frailty
T2	multi-frailty	0.168	0.081	0.272	0.480
	severe socially frailty	0.015	0.621	0.039	0.325
	mild socially frailty	0.105	0.060	0.220	0.614
	relatively healthy frailty	0.071	0.061	0.114	0.754

**Table 4 behavsci-13-00723-t004:** Transition probability of profiles of frailty at T3 and T4.

		T4			
		Multi-Frailty	Severe Socially Frailty	Mild Socially Frailty	Relatively Healthy Frailty
T3	multi-frailty	0.187	0.056	0.207	0.550
	severe socially frailty	0.012	0.747	0.031	0.210
	mild socially frailty	0.105	0.060	0.220	0.614
	relatively healthy frailty	0.054	0.066	0.110	0.770

## Data Availability

The data were obtained from the Chinese Longitudinal Healthy Longevity Survey (CLHLS), which is a public available database. The data are not publicly available due to privacy restrictions.

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
