# Peer review of "Can the Frailty of Older Adults in China Change? Evidence from a Random-Intercept Latent Transition Profile Analysis"

_behavsci, 2023, doi:10.3390/bs13090723_

Round 1
Reviewer 1 Report
Dear Author,
The present research article, entitled “Can the Frailty of Older Adults Change? Evidence from Random-Intercept Latent Transition Profile Analysis”, aims to determine the frailty status and category of older adults according to the physical, psychological, social and cognitive function domains, and on this basis to investigate the transition of the frailty state of older adults.
The main strength of this manuscript is that it addresses a relevant and timely topic by reporting that the frailty status of older adults is heterogeneous and analyzing the maintenance and transitions of frailty status over time.
In general, I believe that the topic and approach of this article is timely and of interest to the readers of Behavioral Sciences. However, I believe that some issues should be included to improve the quality of the manuscript.
1. The introduction is quite complete, but the authors are invited to complete the research hypotheses and expected outcomes of their research.
2. In line 139 (fourth line of the introduction) the country in which the research was carried out is referred to as "my country". However, it is especially important that the wording is formal and that impersonal verbs are used, without using the first person singular or plural.
3. Another important aspect is to put the citations of the instruments used in the text, in the place where they are mentioned for the first time. Although the instruments are mentioned in the appendices, it is necessary for readers to have the citation available so as not to detract from the credibility of the document and, also to be able to use it in possible replications of the research.
4. It is necessary to put below Table 1 (in Note) the meaning of the abbreviations used.
5. It is advisable to justify why frailty profiles are more focused on the social aspect of the elderly than on the other items.
6. The references are not entirely in accordance with the journal's indications. It is recommended to review the "Instructions for Authors" and the reference format of "Behavioral Sciences".
I believe that the manuscript may have an important value in determining frailty in older adults and, consequently, establishing appropriate measures to prevent and intervene in frailty.
I declare no conflict of interest regarding this manuscript.
Best regards,
Reviewer 2 Report
The topic chosen for this study is very interesting and of great importance today. However, there are parts that need to be improved, I hope that with my suggestions and comments they can make those improvements.
1. The objective of the study must be clearly defined and accompanied by hypotheses, where it is specified what the researchers hope to find.
2. In the sample the loss of participants is specified for various reasons, it would be advisable to specify the % of each option. In addition to specifying inclusion and exclusion criteria, and the most relevant sociodemographic characteristics.
3. Calculate Cronbach's alpha for all possible measures, and for those that cannot be specified, specify literature where the reliability and validity of the measure has been verified.
4. Add the type of study design that has been used, as well as the participant recruitment procedure.
5. Both the results and the discussion must have the same order of presentation to be able to follow their understanding more easily.
6. Specify in the title or abstract that it is in the Chinese population.
Round 2
Reviewer 2 Report
All suggestions and comments have been taken into account and have been modified in text, as well as the questions asked. Good job.